# Crystal Nucleation and Growth in Cross-Linked Poly(ε-caprolactone) (PCL)

**DOI:** 10.3390/polym13213617

**Published:** 2021-10-20

**Authors:** Timur Mukhametzyanov, Jürn W.P. Schmelzer, Egor Yarko, Albert Abdullin, Marat Ziganshin, Igor Sedov, Christoph Schick

**Affiliations:** 1A. M. Butlerov Chemical Institute, Kazan Federal University, Kremlevskaya 18, 420008 Kazan, Russia; yarkoeg@gmail.com (E.Y.); alb3978@yandex.ru (A.A.); Marat.Ziganshin@kpfu.ru (M.Z.); igor_sedov@inbox.ru (I.S.); 2Institute of Physics and Competence Centre CALOR, University of Rostock, Albert-Einstein-Str. 23-24, 18051 Rostock, Germany; juern-w.schmelzer@uni-rostock.de

**Keywords:** crystallization, nucleation, cross-links, fast scanning calorimetry (FSC), classical nucleation theory (CNT)

## Abstract

The crystal nucleation and overall crystallization kinetics of cross-linked poly(ε-caprolactone) was studied experimentally by fast scanning calorimetry in a wide temperature range. With an increasing degree of cross-linking, both the nucleation and crystallization half-times increase. Concurrently, the glass transition range shifts to higher temperatures. In contrast, the temperatures of the maximum nucleation and the overall crystallization rates remain the same, independent of the degree of cross-linking. The cold crystallization peak temperature generally increases as a function of heating rate, reaching an asymptotic value near the temperature of the maximum growth rate. A theoretical interpretation of these results is given in terms of classical nucleation theory. In addition, it is shown that the average distance between the nearest cross-links is smaller than the estimated lamellae thickness, which indicates the inclusion of cross-links in the crystalline phase of the polymer.

## 1. Introduction

Poly(ε-caprolactone) (PCL) is a biocompatible and biodegradable industrial polymer with production output amounting to tens of thousands of tons every year. The melting temperature of PCL is low (60 °C), and crystallization proceeds relatively rapidly (at its minimum, the half-time of crystallization is in the order of 0.1 s). The range of current and possible applications of PCL includes rapid prototyping, biomedical products such as sutures and dental splints, controllable drug-release systems, and others [1,2,3]. The properties of PCL can be further tuned to better suit the relevant application by modifying its chemical structure. One option in this direction is creating cross-links between polymer chains of PCL with a suitable cross-linking agent. Such cross-linked polymers based on PCL were shown to have greater mechanical strength and possess a two-way shape memory effect which is not exhibited by the PCL itself [4,5,6,7].

Crosslinking influences the crystallization behavior of the polymer. Under non-isothermal conditions, the crystallization temperature of PCL changes due to cross-linking [8,9]. Optimizing the processing of such modified polymers requires knowledge of their crystallization behavior under different conditions. Notably, the high cooling rates realizable in modern production techniques emphasize the need for assessing the nucleation and crystal growth rate in a wide temperature range, including the deeply supercooled state. At deep supercooling homogeneous crystal nucleation becomes the dominant nucleation process, following the predictions of classical nucleation theory (CNT) [10]. 

The development of fast scanning calorimetry (FSC) [11] has opened the doors to investigate the nucleation and crystallization process of fast crystallizing polymers at low temperatures and deep supercooling conditions [12,13,14,15]. These investigations have provided experimental data on homogeneous nucleation kinetics in polymers and demonstrated that CNT is applicable for describing homogeneous crystal nucleation in polymers. Therefore, combining FSC and CNT should better allow understanding homogeneous nucleation in cross-linked PCL as well. It may be possible to clarify if cross-links are incorporated in the crystal lattice or how cross-links influence the nucleation kinetics. In the present paper, we will use standard approaches of CNT in treating crystallization to perform a theoretical analysis of the obtained experimental data.

However, employing the basic ideas and methods of CNT in polymer crystallization, some specific features must be appropriately accounted for in the theoretical treatment. In particular, non-crystallizable units in the polymer molecule can prevent chains from correct packing. For example, introducing D-isomer units into poly(l-lactic acid) (PLLA) chains reduces polymer crystallinity and slows down both nucleation and crystallization [16]. Crosslinking may act as another sterical hindrance for the chain packing. The spatial density of cross-links introduces another length scale that can affect nucleation and crystallization processes. Comparing nucleation and crystallization kinetics of differently cross-linked polymers to a non-cross-linked sample, a more detailed and correct understanding of structure formation in polymer melts can be, consequently, achieved.

Furthermore, cross-linking should restrict molecular mobility to a greater extent than the inclusion of non-crystallizing units in linear polymer chains. The viscosity of the polymer increases with increasing cross-linking [17]. The cross-linking of PCL has a strong effect on the rheological properties of the polymer, and already at a relatively low cross-link density, the behavior of the polymer changes from viscous to elastic [8]. These variations affect crystal nucleation and growth via variations of the kinetic coefficients in the expressions for the nucleation and growth rates. On the other hand, they may also result in a more pronounced effect of elastic stresses in the work of critical cluster formation, and the driving force for crystal growth resulting in an additional impact on structure formation processes in the polymers analyzed [18], in particular, a decrease in both nucleation and growth rates. Further, at temperatures below the glass transition range, the polymer may not reach the metastable equilibrium state in cooling prior to crystal nucleation. This factor may lead to an additional decrease in the nucleation rate in cooling and support nucleation in heating [19]. In parallel with the change in viscosity, an increase in the glass transition temperature and a slowing down of relaxation processes are expected. It is worthwhile to note that, both glass transition temperature and enthalpy relaxation kinetics of PLLA are not affected by D-isomer’s introduction [20], while the nucleation kinetics and crystal growth show a strong dependency, thus providing a good comparison.

Recently, we have reported [9] the effect of cross-linking density in poly(ε-caprolactone) (PCL) on the non-isothermal crystallization rate of this polymer. The samples of PCL at different degrees of cross-linking were melted and then cooled at different cooling rates; the crystallinity of the sample after this treatment was determined by measuring the fusion enthalpy at subsequent heating. It was found that cross-linking slows down crystallization. The cooling rate at which half of the maximum possible crystallinity of the sample is reached inversely correlates with the cross-link density. Cross-linking modifies the chemical structure of the polymer but keeps most of the monomeric units unchanged. With this, it is an interesting model system to study the influence of the cross-link induced changes in viscosity [21] on crystal nucleation and growth kinetics.

In the present work, we extend the previous study [9] by analyzing crystallization at isothermal conditions down to and below the glass transition temperature. Particularly, we investigate the effect of cross-link density on crystal nucleation and crystallization rates of PCL, as well as the glass transition temperature and relaxation kinetics. In advancing the theoretical analysis, we will analyze which factors employing and going beyond standard CNT are the basic ones determining the effect of cross-linking on crystallization in the polymer analyzed.

## 2. Materials and Methods

Samples of cross-linked PCL were prepared by heating commercial PCL (Aldrich, St. Louis, MO, USA, average *M*_n_ = 45,000 g·mol^−1^, density ρ = 1.142 g·cm^−3^) with different amounts of radical initiator benzoyl peroxide (BPO) (Aldrich, St. Louis, MO, USA, 75% water stabilized), as described previously [4]. The spatial density of cross-links for each sample was determined using an equilibrium swelling method [9], Table 1.

Fast scanning calorimetry experiments were performed using a Flash DSC 1 (Mettler Toledo, Greifensee, Switzerland) with a UFS1 sensor (Mettler Toledo, Greifensee, Switzerland). In every experiment, 10–50 ng of the specimen was placed on the chip sensor, heated up to 150 °C, and cooled down to melt the sample and achieve better thermal contact with the chip sensor before performing measurements.

Tammann’s two-stage nucleation and growth approach, with a non-isothermal growth stage [12,22,23], was employed to investigate the kinetics of crystal nucleation and crystallization of neat and cross-linked PCL. The temperature program includes melting with consecutive fast cooling to an annealing temperature, allowing crystal nucleation and, in some cases, also crystallization. The chosen cooling rate prevents the formation of crystal nuclei before reaching the annealing temperature. The duration and the temperature of the annealing step are varied. After annealing, the sample is cooled at 5000 K/s to −80 °C and then heated to 90 °C (analysis scan). See Figure 1 for details of the temperature program. 

Depending on the annealing conditions, the thermal effects of enthalpy relaxation at the glass transition, cold crystallization (the growth stage in Tammann’s scheme), and melting occur. Because of the decreasing crystallization rates with increasing cross-link density [9], for each sample, a specific heating rate of the analysis scan was chosen. The selected heating rates were so fast that no, or only minor, crystallization occurs for samples not containing homogeneous nuclei. At the same time, the heating rate must be slow enough that for samples containing significant amounts of homogeneous nuclei, crystallization is easily detectable. This way, for all samples containing homogeneously formed crystal nuclei, cold crystallization was observed, independent of the cross-link density dependent crystallization rate.

The optimal heating rate for the analysis scans for the cross-linked PCL samples must be chosen depending on the crystallization rate of the particular polymer. Too slow heating will promote crystallization of the polymer during the analysis scan originating from a few heterogeneous nuclei. At the same time, too fast heating causes a broadening of the cold crystallization peak or even disappearance of cold crystallization and makes an independent evaluation of cold-crystallization and melting enthalpies impossible. As shown previously [9], the crystallization rate of the cross-linked PCL slows down with an increasing degree of cross-linking. Thus, the optimal heating rate depends on the cross-link density. The chosen heating rates were 5000 K/s for pure PCL, 3000 K/s for cross-linked PCL with 3% BPO, 1000 K/s for cross-linked PCL with 5% BPO, and 500 K/s for cross-linked PCL with 10% BPO.

Atomic force microscopy (AFM) images of the samples after crystallization at slow cooling were collected in tapping mode in air using an atomic force microscope Titanium (LLC “NT-MDT”, Moscow, Russia). The instrument is equipped with a multiprobe revolution cartridge of CNG cantilevers (LLC “NT-MDT”, Moscow, Russia) with force constants ranging from 3 to 10 N m^−1^ and resonance frequencies from 120 to 150 kHz. The scan frequency was 1 Hz. The crystalline samples were prepared by melting the polymers between glass slides and allowing the melt to slowly cool in air, after which the samples were cut with a microtome. All images were obtained at 25 °C.

## 3. Results of the Experiments

Figure 2 presents examples of heating scans recorded for the neat PCL sample after annealing at −50 °C. Similar curves were obtained for annealing temperatures from −65 °C up to 20 °C for all cross-linked PCL samples.

At short annealing times, no significant thermal effects are present. With increasing annealing time, a cold-crystallization peak develops, arrow ***b***, indicating the presence of an increasing number of crystal nuclei that can grow to crystals in the temperature range of sufficiently high growth rate [12]. At higher temperatures, the corresponding melting endotherm appears. The position of this endotherm, see arrow ***a***, is essentially independent of the annealing conditions. It is determined by the time available for a melting-recrystallization process on heating [24]. For annealing times above 5 s, a low-temperature endothermic peak develops. It corresponds to the melting of small crystals already grown at the annealing temperature. After this initial melting, the PCL re-crystallizes and re-melts continuously until the final melting occurs, arrow ***a***. 

The values of the cold crystallization enthalpy and overall latent heat (sum of cold-crystallization and all melting enthalpies) were determined as illustrated in Figure 3. Cold crystallization enthalpy corresponds to the area of peak 1, integrated with a linear baseline between points ***a*** and ***b***. The overall latent heat is the integral, employing a linear baseline between points ***a*** and ***c***. 

It can be seen from Figure 2 that, with a longer annealing time, an endothermal peak appears between −30 and 0 °C, indicated by the arrow ***c***. This peak corresponds to the melting of tiny polymer crystals which have grown during the extended annealing. However, the peak may not represent the entire melting endotherm, as the developed melt immediately re-crystallizes to form more stable crystals. The latter effect is exothermal and, thus, partially compensates for the low-temperature melting effect. The re-crystallized crystals finally melt during the high-temperature endotherm. This is illustrated in Figure 3b; the red curve represents the low-temperature melting of the tiny polymer crystals, the blue curve represents the exotherm of recrystallization or cold-crystallization, and the magenta curve corresponds to the melting endotherm of the re-crystallized polymer. The black curve corresponds to the overall effect, which is the sum of the heat flows of the individual processes. The shaded area corresponds to the overall latent heat, which is not affected by the melting-recrystallization process since only the sum of the different heat flows is measured. 

The dependencies of cold-crystallization enthalpy and overall latent heat on annealing time for different annealing temperatures are shown for all studied PCL samples in Figure 4.

As shown in Figure 4, the cold-crystallization enthalpy values have an extremum, while the overall latent heat follows a sigmoidal curve. The initial growth of the cold-crystallization enthalpy is a result of the development of crystal nuclei during annealing. At the same time, the values of the overall latent heat remain close to zero until cold-crystallization reaches the maximum since the formation, and, consequently, the “disordering” of the nuclei does not yield any measurable heat effect.

Further annealing leads to crystal growth during annealing even at the lowest annealing temperatures. Thus, the increasing volume occupied by these crystals does not contribute to the cold-crystallization on heating anymore, and the absolute value of the cold-crystallization enthalpy decreases. At the same time, the overall latent heat starts to deviate from zero and grows continuously. Parameterizing the time dependencies of the cold-crystallization enthalpy and the overall latent heat allows comparing nucleation and crystallization rates between the samples with different cross-link densities. Therefore, the time dependencies of the overall latent heat ($\Delta H_{heating}$) were fitted with Equation (1), following the procedure described in [12]:(1)$$\Delta H_{heating}=\Delta H_{C\infty }\left\{1-\exp{\left(-\ln2\left(\frac{t}{\tau_{c}}\right)\right)}^{n_{c}}\right\}+A_{2}\ln\left(\frac{t}{\tau_{c}}\right)\cdot \frac{1}{2}\cdot \left(\frac{\left|t-\tau_{c}\right|}{t-\tau_{c}}+1\right)$$

The first term of this equation corresponds to the standard Johnson-Mehl-Avrami-Kolmogorov (JMAK) description of overall crystallization. Here, $\Delta H_{C\infty }$ is the enthalpy of melting at the final stage of primary crystallization, $\tau_{c}$ is the halftime of crystallization, and $n_{c}$ is the Avrami exponent of crystallization. The second term describes secondary crystallization, and its treatment is based on the assumption of a linear increase in melting enthalpy with the logarithm of time. Figure 5 shows an example of the fit of the overall latent heat of the pure PCL sample after annealing at –45 and –60 °C with Equation (1). The corresponding fitting parameters are: at –45 °C $\Delta H_{C\infty }$ = (3.1 ± 0.1) × 10^−4^ J, $\tau_{c}$ = 5.3 ± 0.3 s, $n_{c}$ = 2.5 ± 0.4, *A*_2_ = (1.0 ± 0.2) × 10^−5^ J; at –60 °C $\Delta H_{C\infty }$ = (4.4 ± 0.2) × 10^−4^ J, $\tau_{c}$ = 119 ± 13 s, $n_{c}$ = 1.0 ± 0.1, *A*_2_ = 0 J.

The parametrization of the cold crystallization time-dependencies was carried out as follows. For each cross-link density of PCL, a set of cold-crystallization curves was obtained. Among each set, the curve with the highest absolute cold-crystallization enthalpy value was chosen. The data points after the extremum of the cold-crystallization curve were discarded, and the remaining data points were fitted with the JMAK-equation:(2)$$\Delta H_{CC}=\Delta H_{CC\infty }\left\{1-\exp{\left(-\ln2\left(\frac{t}{\tau_{n}}\right)\right)}^{n_{n}}\right\}$$
where $\Delta H_{CC\infty }$ is the maximum of the cold crystallization enthalpy of the considered polymer, $\tau_{n}$ is the halftime of nucleation, $n_{n}$ is the Avrami exponent for the nucleation process. For the rest of the curves $\Delta H_{CC\infty }$ and $n_{n}$ were kept fixed, and only nucleation halftime was allowed to fit. Again, the points after the extremum of the curve were discarded since they are strongly disturbed by the growth of crystals.

An example of the fit for the time dependence of the cold crystallization enthalpy of PCL with 5% BPO after annealing at −45 and −60 °C with Equation (2) is also shown in Figure 5. The corresponding fitting parameters are: at −45°C $\Delta H_{CC\infty }$ = −(2.4 ± 0.1)·10^−4^ J, $\tau_{n}$ = 5.2 ± 0.3 s, *n*_n_ = 1.5 ± 0.1; at −60 °C $\Delta H_{CC\infty }$ = −(2.4 ± 0.1)·10^−4^ J, $\tau_{n}$ = 2.8 ± 0.1 s, *n*_*n*_ = 1.5 ± 0.1.

The temperature dependence of nucleation and crystallization halftimes determined using the procedure described above is presented in Figure 6.

For all samples, the nucleation and crystallization halftime minima appear at about −55 °C and 0 °C, respectively. These minima are equivalent to the maxima of homogeneous nucleation and overall crystallization rates. The independence of the position of the maxima of homogeneous nucleation rates on cross-link density follows directly from the experimental data. The corresponding half-times of cold crystallization are not disturbed by other effects. The position of the maximum of the overall crystallization rate at about 0 °C depends on the interplay between heterogeneous nucleation kinetics and the temperature dependence of the crystal growth rate. With the data shown in Figure 6, it is impossible to decide if the heterogeneous nucleation rate or growth rate causes the stable position of the minimum crystallization halftime. There are even arguments that the temperature position of the maximum of the overall crystallization rate may be close to the maximum of the crystal growth rate [25]. 

Disentangling the influence of heterogenous nucleation kinetics and growth rate requires some general considerations. Starting from the Johnson–Mehl–Avrami–Kolmogorov (JMAK) approach [18,26], the temperature of the maximum growth rate becomes accessible [25]. On heating an amorphous but nuclei containing sample, cold crystallization is observed, see Figure 2. The cold crystallization peak maximum shifts to higher temperatures with increasing heating rate before the peak eventually vanishes. From the theoretical analysis, it follows that the position of the peak before it vanishes approaches the temperature of the maximum crystal growth rate. The heating rate at which the cold crystallization temperature reaches a plateau depends on the nucleation state of the sample. Figure 7 shows the heating rate dependence of the cold crystallization peak temperature for three different samples of the cross-linked PCL with 5% BPO. The blue curve was obtained when the sample was heated quickly (1 s isotherm at −80 °C) after rapid cooling at 5000 K/s to −80 °C. Red and black curves were obtained from two distinct samples (with different mass and geometry) after annealing (nucleating) the samples at −60 °C for 20 s (1000 K/s cooling rate). As can be seen from Figure 7, depending on the sample history, distinct dependencies of the cold crystallization peak temperature on the heating rate are obtained, but the asymptotic temperatures of all curves are nearly the same. The heating rate when the non-nucleated sample reaches the asymptotic cold crystallization peak temperature is much lower than that for the nucleated samples. It indicates that the critical heating rate for preventing nucleation is much lower than the critical heating rate for preventing crystal growth from preexisiting nuclei. In both cases the temperature of maximum growth rate provides the limit for observable crystal growth. 

The data points from Figure 7 are fitted by
(3)$$T_{max,CC}\left(\beta \right)=T_{max,CC}^{lim}+\left(T_{max,CC}^{0}-T_{max,CC}^{lim}\right)\cdot \exp\left(-A\cdot \beta \right)$$
where $T_{max,CC}\left(\beta \right)$ is the cold crystallization peak temperature at heating rate β, $T_{max,CC}^{lim}$ is the asymptotic value of the cold crystallization peak temperature, $T_{max,CC}^{0}$ is the cold crystallization peak temperature at a heating rate approaching zero, *A* is an empirical constant.

The asymptotic value, $T_{max,CC}^{lim}$, of the fit function represents a temperature close to the maximum of the crystal growth rate and is presented for all PCL samples as a function of the cross-link density in Figure 8.

The asymptotic value of the cold crystallization peak temperature is located, independent of cross-link density, at about 5 °C, which is in good agreement with the maximum position of the overall crystallization rate in Figure 6. This way, the temperature of the maximum of the overall crystallization rate can be interpreted as being caused by the maximum of the crystal growth rate and only marginally influenced by heterogeneous nucleation kinetics. Interestingly, the maximum of the overall crystallization rate in Figure 6 and the temperature of maximum growth rate from Figure 8 are independent of the cross-link density.

The classical nucleation theory describes both nucleation and crystal growth rate as dependent on diffusivity [25]. The diffusion coefficient is commonly linked to viscosity and segmental mobility in polymers [18]. Therefore, the glass transition temperature of the different PCL samples was determined. Although the heating scans performed during the annealing experiments did show glass transition, the temperature programs had to be optimized to provide a clear comparison between the different PCL samples in this respect. Thus, a separate set of experiments was performed to check the effect of cross-links on the glass transition of PCL. The glass transition temperature at heating rate 1000 K/s is almost the same in neat PCL, cross-linked PCL with 3% and 5% BPO (glass transition midpoints −65.6, −65, and −64.6 °C, respectively), but is notably higher for cross-linked PCL with 10% BPO (glass transition midpoint −54 °C). 

Summarizing briefly the experimental data, we come to the following conclusions: With increasing degree of cross-linking, both the nucleation and crystallization half-times increase. In parallel, as shown in Figure 9, the glass transition range shifts to higher temperatures. In contrast, the temperatures of the maximum nucleation and the overall crystallization rates remain the same independent of the degree of cross-linking. The cold crystallization peak temperature increases generally as a function of heating rate reaching an asymptotic value near to the temperature of the maximum growth rate. A theoretical interpretation of these results is given in terms of classical nucleation theory in the subsequent section. In addition, it is shown there that the average distance between the nearest cross-links is smaller than the estimated lamellae thickness, which indicates the inclusion of cross-links in the crystalline phase of the polymer.

## 4. Theoretical Analysis

### 4.1. Basic Theoretical Relations

Our analysis is performed in terms of the classical theories of nucleation and growth [18], utilizing largely the results obtained by us in [25]. In terms of CNT, the steady-state nucleation rate, *J*, is expressed as
(4)$$J=c\sqrt{\frac{\sigma }{k_{B}T}}\left(\frac{D}{d_{0}}\right)\exp\left(-\frac{W_{c}}{k_{B}T}\right)$$
where, *σ* is the surface tension, *k_B_* the Boltzmann constant, *T* the absolute temperature, *D* the effective diffusion coefficient governing the processes of aggregation of ambient phase particles to crystal clusters, and *d*_0_ is a characteristic size parameter that is determined by the particle number density, *c*, of the basic units of the ambient phase (*c* = 1/*d*_0_^3^).

In the case of homogeneous nucleation of spherical nuclei, the size of the critical cluster, *R_c_*, its surface area, *A_c_*, and the work of critical cluster formation, *W_c_*, are given by the following relations:(5)$$R_{c}≅\frac{2\sigma }{c\Delta \mu },\;W_{c}=\frac{1}{3}\sigma A_{c}=\frac{16\pi }{3}\frac{\sigma^{3}}{{\left(c\Delta \mu \right)}^{2}},\;A_{c}=4\pi R_{c}^{2}$$
where, ∆*µ* is the difference in the chemical potential per particle in the liquid and the crystal. Consequently, the volume of a critical cluster is given by:(6)$$v_{c}=\frac{4\pi }{3}R_{c}^{3}=\frac{4\pi }{3}{\left(\frac{2\sigma }{c\Delta \mu }\right)}^{3}$$

In its simplest form, the thermodynamic driving force, ∆*µ*, for nucleation and growth can be expressed as:(7)$$\Delta \mu =q\left(1-\frac{T}{T_{m}}\right),\;q=T_{m}\Delta s_{m}$$
where *q* (*q* > 0) is the latent heat of crystallization per particle and ∆*s_m_* is the melting entropy per particle at the equilibrium melting (or liquidus) temperature, *T_m_*. The surface tension, *σ*, is estimated via the Stefan–Skapski–Turnbull relation [18] as:(8)$$\sigma =\alpha \frac{q}{v^{2/3}},\;v=\frac{1}{c}=d_{0}^{3}$$
where the surface tension is assumed to be equal to its value for an equilibrium coexistence of liquid and crystal at planar interfaces at the melting temperature, *T_m_*. A temperature or size dependence of the surface tension can be introduced as described in [27]. Employing Equation (8), surface effects enter the description via the melting entropy and the parameter *α*. For different systems, this parameter was found to have values commonly in the range 0.3 *< α <* 0.6.

In such terms, the work of critical cluster formation can be written in the form:(9)$$\frac{W_{c}}{k_{B}T}=\frac{\Omega }{{\left(1-\frac{T}{T_{m}}\right)}^{2}},\;\Omega =\frac{16\pi \alpha^{3}\Phi }{3}\left(\frac{q}{k_{B}T_{m}}\right)$$

This relation accounts for both homogeneous (*Φ* = 1) and heterogeneous (*Φ* < 1) nucleation. Employing these notations, the steady-state nucleation rate is given by:(10)$$J≅c\sqrt{\frac{\alpha q}{k_{B}T_{m}}}\left(\frac{D}{d_{0}^{2}}\right)\exp\left(-\frac{\Omega }{\frac{T}{T_{m}}{\left(1-\frac{T}{T_{m}}\right)}^{2}}\right)$$

In this equation, only one parameter reflects the bulk properties of the substance under consideration, the ratio of the latent heat, *q*(*T_m_*), divided by the characteristic thermal energy, *k_B_T_m_*.

For the macroscopic linear growth rate, *u*, we use the commonly employed relation [28]
(11)$$u=f\frac{D}{4d_{0}}\left[1-\exp\left(-\frac{\Delta \mu }{k_{B}T}\right)\right]$$
where *f* ≤ 1 is a parameter that has different values for different modes of growth. We suppose that the kinetics of aggregation is the same for both nucleation and growth and is governed by a diffusion coefficient, *D*, which can be written as:(12)$$D=D_{0}\exp\left(-\frac{E_{D}}{k_{B}T}\right)$$

The activation energy for diffusion, *E_D_* = *E_D_*(*T*), depends on temperature, pressure, and composition. Pressure and composition are assumed to be constant and not affected by the phase formation processes considered.

In applications, the diffusion coefficient, *D*, governing nucleation and growth is usually not known and is therefore frequently estimated via the Stokes–Einstein–Eyring (SEE) relation [18]:(13)$$D≅D_{\eta }=\gamma \frac{k_{B}T}{d_{0}\eta }$$

This equation allows one to replace the diffusion coefficient, *D*, by the Newtonian viscosity, *η*. The parameter *γ* is a constant. Its value depends on the theoretical concepts and approximations employed in the derivation, and on the way of specification of the size parameter, *d*_0_, of the basic molecular units of the melt utilized. Provided that Equation (13) holds, *D* can be replaced by the viscosity, which is described by a relation similar to Equation (12):(14)$$\eta =\eta_{0}\;\exp\left(\frac{E_{\eta }}{k_{B}T}\right)$$
with
*E_D_*(*T*) = *E_η_*(*T*)(15)

The SEE-relation is commonly applicable only above a certain decoupling temperature, *T_d_*
≅ (1.1 − 1.2)*T_g_*. However, qualitatively, it correctly describes the correlation between diffusion and viscosity also below *T_g_*. By this reason, we will employ the SEE-relation and its consequences for estimates in the whole temperature range.

### 4.2. Half-Times of Nucleation and Overall Crystallization

The nucleation half-time is the time required to fill the volume of the liquids phase to a certain degree with the newly evolving crystal phase accounting only for nucleation. The nucleation rate gives the number of critical crystals ($N_{c}$) formed per unit time in a unit volume (e.g., 1 m^3^). Then, the nucleation half-time is given by
(16)$$N_{c}v_{c}\;=Jv_{c}\tau_{n}\;=0.5$$
or
(17)$$\tau_{n}=\frac{0.5}{\frac{4\pi }{3}{\left(\frac{2\sigma }{c\Delta \mu }\right)}^{3}}\frac{1}{J}=\frac{1.5}{4\pi }{\left(\frac{c\Delta \mu }{2\sigma }\right)}^{3}\frac{1}{J}$$
if the volume of the crystal phase is equal to one half of the volume of the liquid. The subsequently performed theoretical considerations are independent of this assumption and valid for any choice of this ratio. With Equations (4) and (12), we can rewrite the latter relation as:(18)$$\tau_{n}\propto {\left(\frac{c\Delta \mu }{2\sigma }\right)}^{3}\exp\left(\frac{W_{c}+E_{D}}{k_{B}T}\right)$$

According to the Johnson–Mehl–Avrami–Kolmogorov approach [18,25,26,29] to the determination of the volume fraction, *α*, of the crystalline phase as a function of time at isothermal process conditions, the maximum rate, (*dα*(*t*,*T*)*/dt*), of overall crystallization as a function of temperature (at any given value of crystal volume fraction, *α*, and time, *t*) is determined by the product *J*(*T*)*u^n^*(*T*):(19)$$\alpha \left(t,T\right)=1-\exp\left(-\frac{\omega }{n+1}Ju^{n}t^{n+1}\right)$$
where, *n* is the number of independent directions of growth and *ω* is a shape factor. For the half-time of crystallization at isothermal conditions, we obtain:(20)$$\tau_{C}=\sqrt[{n+1}]{0.7\frac{n-1}{\omega }}\frac{1}{\sqrt[{n+1}]{Ju^{n}}}$$

We obtain similarly to Equation (18) the following estimate:(21)$$\tau_{C}\propto {\left(\exp\left(\frac{W_{c}+E_{D}}{k_{B}T}\right)\right)}^{1/\left(n+1\right)}{\left(\frac{\exp\left(\frac{E_{D}}{k_{B}T}\right)}{1-\exp\left(-\frac{\Delta \mu }{k_{B}T}\right)}\right)}^{n/\left(n+1\right)}$$

Utilizing Equations (18) and (21), the increase in the half-times of nucleation and crystallization with the degree of cross-link densities (shown in Figure 6) can be interpreted, in general, via the dependence of these time-scales on heat of melting, the melting temperature, the work of critical cluster formation and the activation energy for diffusion.

The increase in the activation energy for diffusion is confirmed independently by the increase in the viscosity and the glass transition temperature with the increase in cross-link density. The increase in the viscosity may result in an increase in the work of critical cluster formation due to the evolution of elastic stress effects in nucleation [29]. Latter effect is not of significance for the change of the crystallization time, since at higher temperatures elastic stress effects cannot have any effect on nucleation. Quite remarkably, such general behavior of the nucleation and crystallization half-times is accompanied by nearly constant temperatures both of the minimum of these characteristic time-scales. This independence on cross-link density can be theoretically interpreted in the following way.

As shown in [25], the temperature $T_{max}^{\left(nucl\right)}$ of the maximum nucleation rate or the minimum of the half-time of nucleation corresponds to the minimum of the ratio (*W_c_* + *E_D_*)/*k_B_T*. It is given generally by:(22)$$\frac{T_{max}^{\left(nucl\right)}}{T_{m}}=\frac{1}{T_{m}}{\left(\frac{W_{c}\left(T\right)+E_{D}}{\frac{d\left(W_{c}\left(T\right)+E_{D}\right)}{dT}}\right)|}_{T=T_{max}^{\left(nucl\right)}}$$

Employing the approximations leading to Equation (9), this relation can be transformed to:(23)$$\frac{T_{max}^{\left(nucl\right)}}{T_{m}}={\left(\frac{W_{c}\left(T\right)+\left(E_{D}-T\frac{dE_{D}}{dT}\right)}{3W_{c}\left(T\right)+\left(E_{D}-T\frac{dE_{D}}{dT}\right)}\right)|}_{T=T_{max}^{\left(nucl\right)}}$$

As a rule, one can expect (c.f. Equation (2.80) in [18]) that the inequality
(24)$$\frac{dE_{D}}{dT}\leq \;0$$
holds. Changes of the work of critical cluster formation and the activation energy for diffusion result both in variations of the numerator and the denominator in Equation (23) partly or completely removing any effect of cross-link density on the half-time of nucleation.

As also shown in [25], employing Equation (7), the generally valid result (Equations (15), (33) and (34) in [25])
(25)$${\exp\left(\frac{\Delta \mu }{k_{B}T}\right)|}_{T=T_{max}^{\left(growth\right)}}=1+{\frac{q}{E_{D}^{\left(eff\right)}}|}_{T=T_{max}^{\left(growth\right)}},\;E_{D}^{\left(eff\right)}\left(T\right)=\left(E_{D}-T\frac{dE_{D}\left(T\right)}{dT}\right)$$
for the temperature of the maximum growth rate, $T_{max}^{\left(growth\right)}$, can be simplified to
(26)$$\frac{T_{max}^{\left(growth\right)}}{T_{m}}={\frac{1}{1+\frac{k_{B}T_{m}}{q}\ln\left(1+\frac{q}{E_{D}^{\left(eff\right)}}\right)}|}_{T=T_{max}^{\left(growth\right)}}$$

It is evident that cross-link densities affect the location of the maximum growth rate mainly via variations of the melting entropy and melting temperature.

Further, it is shown there that the maximum of the overall-crystallization rate or the minimum of the half-time of crystallization is located at temperatures, $T_{max}^{\left(overall\right)}$, in between $T_{max}^{\left(nucl\right)}$ and $T_{max}^{\left(growth\right)}$, i.e., $T_{max}^{\left(nucl\right)}\leq T_{max}^{\left(overall\right)}\leq T_{max}^{\left(growth\right)}$. The temperature $T_{max}^{\left(overall\right)}\;$is determined by the following relation:(27)$$\frac{T_{max}^{\left(overall\right)}}{T_{m}}={\frac{1-\Upsilon }{3-\Upsilon }|}_{T=T_{max}^{\left(overall\right)}}$$
(28)$$\Upsilon \left(T_{max}^{\left(overall\right)}\right)=\left(n+1\right)\frac{E_{D}^{\left(eff\right)}\left(T_{max}^{\left(overall\right)}\right)}{W_{C}\left(T_{max}^{\left(overall\right)}\right)}\;\;\;\;\;\;\;\;\;\;\;\;\;\;\;\times {\left\{\frac{nq}{\left(n+1\right)E_{D}^{\left(eff\right)}\left(\exp\left(\frac{\Delta \mu }{k_{B}T}\right)-1\right)}-1\right\}|}_{T=T_{max}^{\left(overall\right)}}$$

At this temperature, the driving force of crystallization is a relatively small quantity and we can approximate Equation (28) via
(29)$$\Upsilon \left(T_{max}^{\left(overall\right)}\right)={\frac{nq}{W_{C}\left(\exp\left(\frac{\Delta \mu }{k_{B}T}\right)-1\right)}|}_{T=T_{max}^{\left(overall\right)}}$$

Employing such approximation, $T_{max}^{\left(overall\right)}\;$becomes independent on the activation energy of diffusion and is affected only by variations of the work of critical cluster formation and the heat of melting. However, again, Υ is present in both the numerator and the denominator in Equation (27) resulting in a wide independence of the temperature of the minimum of the half-time of crystallization on cross-link density.

### 4.3. Dependence of the Cold-Crystallization Peak Temperature on the Heating Rate

The analysis of nucleation-growth processes at changing temperatures is a much more complex problem as compared to the theoretical treatment of this process at isothermal process conditions. In order to describe them appropriately, one could perform numerical computations based on the basic set of kinetic equations describing the evolution of the cluster size distributions function in dependence on temperature and time [18]. However, for a variety of applications, the knowledge of relatively simple analytical expressions is desired, allowing one to determine the temperature of the peak of crystallization as a function of the rate of change of temperature.

Extending previously obtained results, in [28] the average time of formation of the first supercritical nucleus in cooling and heating was specified. These results allow one to determine the time and temperature when the nucleation-growth processes become of importance. The crystallization peak is determined, then, as the result of nucleation and subsequent growth of the supercritical clusters proceeding after the first nucleus has been formed. Different approaches have been developed in the past to describe this process [30,31,32,33,34]. In these attempts, the degree of crystallization is expressed as some function of temperature introducing some activation energy chosen in such a way that the crystallization peaks in heating or cooling are specified more or less correctly.

From a mathematical point of view, the degree of crystallization is described as a function of temperature and time. However, such treatment is not correct.

Indeed, as described, e.g., in [18], the general relation for the change of the degree of crystallization in terms of the JMAK-approach is given by:(30)α(t)=1− exp[−Y (t)]
(31)$$Y\left(t\right)=\omega \overset{t}{\underset{0}{\int }}J\left(t^{\prime }\right)dt^{\prime }{\left(\overset{t}{\underset{0}{\int }}u\left(t^{\prime \prime }\right)dt^{\prime \prime }\right)}^{n}$$
where *α* is the volume fraction of the crystal phase, *J* is the rate of nucleation of supercritical crystallites and *u* their rates of growth, *ω* is a shape factor. In this integration procedure it is assumed that both *α* and *Y* are equal to zero at *t* = 0. We assume here the crystallites to be of spherical shape with a radius, *R*, characterized by a growth rate, *u* = (*dR/dt*), and *ω* = 4*π*/3. In line with latter two relations, the degree of crystallization at some given time is not a function of temperature but a functional of combinations of the nucleation and growth rates. Solving these relations numerically, we can also in terms of the JMAK-treatment determine the dependence of the degree of crystallization on time or temperature including, as a special case, the temperatures of the crystallization peaks. In the present analysis, we will derive simple relations for the dependence of the cold-crystallization peak temperature on the heating rate. In this procedure, certain approximations are required. A comparison of numerical and analytical results will be presented in a forthcoming analysis.

We assume that, in heating, independent of the rate of heating the maximum rate of overall crystallization always corresponds to the same amount of the crystal phase or the same value of the function *Y* (*T_p_*) = constant. Then, for a given value of this parameter, we can evaluate how the peak-temperature *T_p_* depends on the rate of change of heating. Assuming the heating rate to be equal to *q_h_* = *dT/dt >* 0, we may express *Y* as a function of temperature as:(32)$$Y\left(T\right)=\frac{\omega }{q_{h}^{n+1}}\left\{\int_{T_{s}}^{T}J\left(T^{\prime }\right)dT^{\prime }{\left(\int_{T^{\prime }}^{T}u\left(T^{\prime \prime }\right)dT^{\prime \prime }\right)}^{n}\right\}.$$
where *T*_s_ is the temperature at which the heating is started located well-below the temperatures of the maxima of nucleation and growth rates. Quite generally, *J*(*T*) is different from zero only in a small temperature interval around $T=T_{max}^{\left(nucl\right)}$. Additionally, this temperature range is located well below the maximum of the growth rates (see Figure 10). Typical locations of the maximum of the nucleation and growth rates confirming this statement are shown in Figure 10.

For such cases, we may write approximately:(33)$$Y\left(T\right)=\frac{\omega }{q_{h}^{n+1}}\left\{J\left(T_{max}^{\left(nucl\right)}\right)\Delta T_{max}^{\left(nucl\right)}{\left(\overset{T}{\underset{T^{\prime }}{\int }}u\left(T^{\prime \prime }\right)dT^{\prime \prime }\right)}^{n}\right\}$$
(34)$$J\left(T_{max}^{\left(nucl\right)}\right)\Delta T_{max}^{\left(nucl\right)}=\overset{T}{\underset{T_{s}}{\int }}J\left(T^{\prime }\right)dT^{\prime }.$$

The growth rates have non-zero values only at temperatures, *T*, obeying the inequality $T_{max}^{\left(nucl\right)}\ll T_{min}^{\left(growth\right)}\leq T$. Here, $T_{min}^{\left(growth\right)}$ is the lower bound of the temperature interval, where the growth rate becomes appreciable. By this reason, we can reformulate above relation also as
(35)$$Y\left(T\right)=\frac{\omega }{q_{h}^{n+1}}\left\{J\left(T_{max}^{\left(nucl\right)}\right)\Delta T_{max}^{\left(nucl\right)}{\left(\int_{T_{min}^{\left(growth\right)}}^{T}u\left(T^{\prime }\right)dT^{\prime }\right)}^{n}\right\}$$
Assuming, as mentioned, *Y* (*T_p_*) = constant, this relation describes how *T_p_* depends on the heating rate,
(36)$$Y\left(T_{p}\right)=\frac{\omega }{q_{h}^{n+1}}\left\{J\left(T_{max}^{\left(nucl\right)}\right)\Delta T_{max}^{\left(nucl\right)}{\left(\int_{T_{min}^{\left(growth\right)}}^{T_{p}}u\left(T^{\prime }\right)dT^{\prime }\right)}^{n}\right\}\;=constant$$ Accounting for the mathematical identity:(37)$$\int_{a}^{b}y\left(x\right)dx=y\left(\langle x\rangle \right)\left(b-a\right),\;a\leq (\langle x\rangle )\leq b$$
this relation can be rewritten as:(38)$${\left(T_{p}-T_{min}^{\left(growth\right)}\right)}^{n}=C_{1}\frac{q_{h}^{n+1}}{u^{3}\left(\langle T\rangle \right)},\;C_{1}=constant$$
where 〈T〉 obeys the condition $T_{min}^{\left(growth\right)}\leq \langle T\rangle \leq T_{p}$. Additionally, considering u(〈T〉) as nearly constant, we get a simple first estimate for the dependence of the peak temperature on the heating rate. With an increase in *q_h_*, *T_p_* has to increase (becoming nearer to the temperature corresponding to the maximum of the growth rate or even slightly larger) in order to realize the condition given by Equation (36). Note that similar estimates hold, as well as if in the initial cooling process of the sample some clusters are already formed. Their account will merely result in a modification of the constant in Equation (38).

Equation (38) predicts an increase in the peak temperature with increasing heating rate as observed in the experiments and illustrated in Figure 7. As evident from the above derivations, it can be treated as a consequence of nearly constant values of Y. With an increase in the heating rate, higher temperatures have to be approached in order to compensate the variation of the heating rate by the integral term containing the growth rate of the supercritical clusters. However, such a mechanism works only up to temperatures corresponding to the maximum of the growth rate. By this reason, the peak temperature approaches a saturation value near to the temperature of the maximum growth rate. Moreover, a further increase in the rate of change of temperature may result in the disappearance of such peak temperature. These effects are desribed briefly in the subsequent section.

### 4.4. Number of Supercritical Clusters in Dependence on Heating Rate

Provided the cooling rate is higher than the critical cooling rate for crystallization, for the interpretation of the results using above estimates, we have to employ Equation (36). In this approach, it was accounted for that the nucleation rate is essentially different from zero only in a small temperature interval. In the derivation, the transformation
(39)$$\overset{t}{\underset{0}{\int }}J\left(t^{\prime }\right)dt_{\prime }=\frac{1}{q_{h}}\overset{T}{\underset{T_{s}}{\int }}J\left(T^{\prime }\right)dT^{\prime }=\frac{1}{q_{h}}\;J\left(T_{max}^{\left(nucl\right)}\right)\Delta T_{max}^{\left(nucl\right)}$$
was used. Since
(40)$$N_{c}=\overset{t}{\underset{0}{\int }}J\left(t^{\prime }\right)dt^{\prime }$$
is the number of supercritical clusters formed in heating, it follows that the number of critical clusters formed in heating is inversely proportional to the heating rate. This is one of the major reasons why for a non-nucleated sample, a cold crystallization peak can be obtained only below certain maximum values of the heating rate. Note that the existence of supercritical nuclei becomes evident only via their subsequent growth. The characteristic times at which clusters may grow are also determined by the heating rate. Consequently, even if a set of supercritical clusters is available prior to heating, they may not lead to a peak temperature due to a negligible increase in their radii in rapid heating. In such more general situation, we have consequently merely to add to the term *J*Δ*T* in Equation (36) the number of supercritical clusters already present in the system prior to the heating procedure.

### 4.5. Cross-Link Density and Their Effect on the Evolving Crystalline Phase

For chemically modified semi-crystalline polymers, the question arises if the modified units are excluded from the crystals or incorporated. To answer this question, the distance between cross-links is compared with the size of critical clusters or the thickness of crystalline lamellae. According to the previous measurements [9], the cross-linked PCL with 10% BPO has a cross-link density of about 200 mol/m^3^. In the case of an equally spaced distribution of cross-links forming a cubic lattice, the distance *d* between the nearest of them can be calculated as:(41)d=1N·NA3
where *N* is the cross-link density (mol·m^−3^), *N_A_* is Avogadro’s number (mol^−1^), giving the nearest distance between cross-links of about 2 nm.

If we assume a totally random (Poisson) distribution of the cross-links in space, then the average distance 〈d〉 between the nearest cross-links is given by:(42)$$\langle d\rangle =\sqrt[{3}]{\frac{3}{4\pi NN_{A}}}\Gamma \left(\frac{4}{3}\right)$$
where Γ is the gamma-function. For cross-linked PCL with 10% BPO, it results in an average distance of about 1.1 nm. The average distances between the cross-links for the studied PCL samples are presented in Table 2.

Even if the estimates for the distances between cross-links are very rough, it is worth comparing them with the expected sizes of critical crystal nuclei or the lamellae thicknesses of PCL. For neat PCL, the critical cluster radius at –65 °C is ca. 4.8 nm [13], and this value increases with temperature. For the lamellae, the Gibbs–Thomson relation yields a comparable thickness of about 4.5 nm at 30 °C [22]. Assuming that the critical cluster sizes and lamellae thicknesses are similar in cross-linked PCL, the average distance between cross-links is smaller than these sizes, even at the lowest annealing temperatures used in the present work. Consequently, the cross-links are most probably incorporated into the polymer crystals in the studied PCL samples.

Another argument for this conclusion is the retention of comparable crystallinity in the neat and all cross-linked PCL samples. Although the crystallinity of the cross-linked PCL, expressed by the fusion enthalpy of the slowly crystallized sample (conventional DSC), decreases, the degree of crystallinity of cross-linked PCL with 10% BPO is no less than 77% of the neat PCL [9]. Thus, considering that crystallization, in this case, occurs at relatively low supercooling and corresponding critical cluster sizes and lamellae thicknesses must be well above the distances between cross-links, we may conclude that the cross-links are incorporated in the crystal lattice in the case of the cross-linked PCL 

To confirm the inclusion of cross-links into larger-scale crystallites, we have collected AFM images of the samples of neat PCL and cross-linked PCL with 5% BPO, prepared by slow cooling of the melt. The spherulitic crystals with sizes of up to 20 micrometers are visible in both cross-linked and non-cross-linked PCL samples, Figure 11.

On a smaller scale, the internal structure of the crystallites appears to be different (Figure 12), though lamellae-like structures are visible in both samples. 

The consequences of the cross-links for the crystal structure are not yet known in detail. However, the AFM images and the other arguments highlighting the inclusion of cross-links in the crystal nuclei and crystalline lamellae, suggesting that the structure of the PCL crystals is not too much modified by cross-linking.

## 5. Conclusions

The nucleation and crystallization kinetics of cross-linked poly(ε-caprolactone) was investigated in wide temperature ranges by fast scanning calorimetry. The rates of nucleation and the overall crystallization become progressively lower as the cross-linking degree increases. However, the temperatures of the maxima of nucleation and the overall crystallization rates do not depend on the degree of cross-linking. The interpretation of the experimental results is given based on the classical nucleation theory framework. The theoretical analysis confirms the independence of the temperatures of the maxima of nucleation and overall crystallization rate on the degree of cross-linking.

We also demonstrate that the cold crystallization peak temperature increases with increasing heating rate, reaching an asymptotic value near the temperature of the maximum overall growth rate. The limiting value of the cold crystallization peak temperature does not depend on the number of nuclei in the sample and the cross-linking degree. The analytical function for the dependence of the cold crystallization peak temperature on the heating rate is presented.

We have estimated the average distances between the cross-links from the cross-link density of the PCL samples. These distances appear to be smaller than the critical cluster size and the lamellae thicknesses, calculated based on the classical nucleation theory. As the presence of the typical spherulites and lamellae structures in the cross-linked PCL is verified by the AFM images, we can conclude that the cross-links are included in the crystal structure and not confined in the amorphous regions.

## Figures and Tables

**Figure 1 polymers-13-03617-f001:**
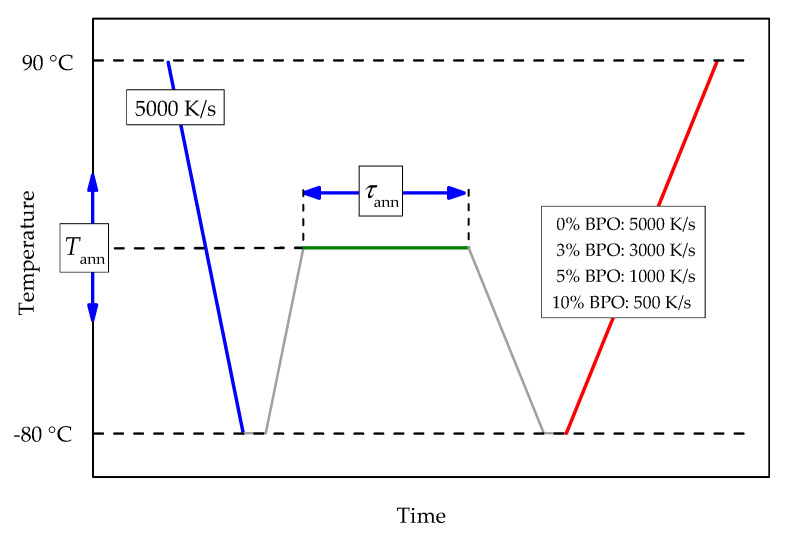
Scheme of the temperature program for the investigation of crystal nucleation and crystallization kinetics of cross-linked PCL. Cooling and heating rates for all steps except the analysis scan are 5000 K/s. The heating rate of the analysis scan depends on the cross-link density of the sample; red line, see the legend. Annealing temperatures (*T*_ann_) are between −65 °C and 20 °C with a 5 °C increment. Annealing times (τ_ann_) are between 0.01 and 5000 s evenly spaced on a logarithmic scale.

**Figure 2 polymers-13-03617-f002:**
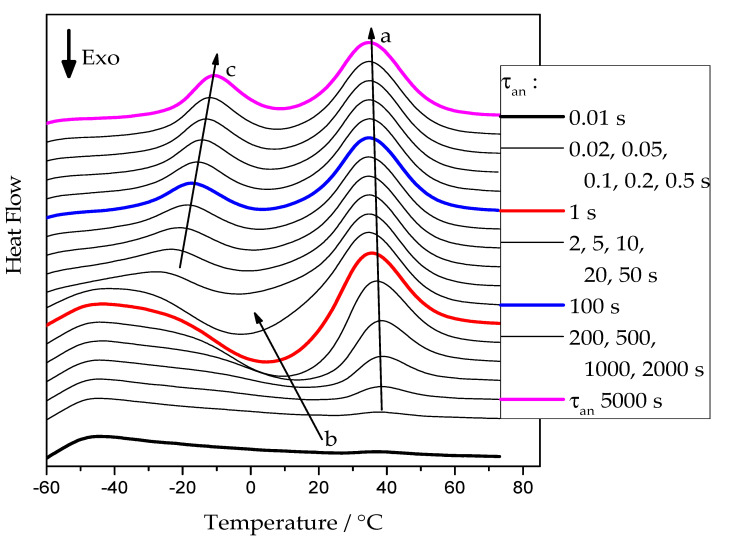
FSC heating curves of pure PCL at 5000 K/s after annealing at −50 °C for various times. Arrows: a—development of the melting peak of the PCL re-crystallized on heating; b—development of the cold-crystallization exotherm; c—development of the melting peak of the tiny polymer crystals grown at the annealing temperature.

**Figure 3 polymers-13-03617-f003:**
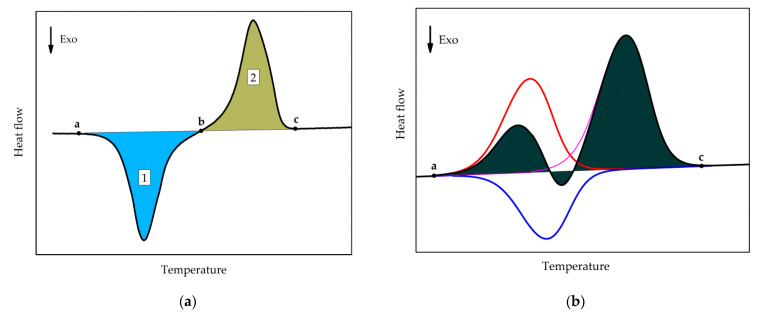
Schematics of (**a**) Determination of the cold-crystallization enthalpy (area of the peak 1) and overall latent heat (sum of the areas of peaks 1 and 2). (**b**) Determination of the overall latent heat (shaded area), red curve: melting of tiny polymer crystals, blue curve: recrystallization effect, magenta curve: melting of the re-crystallized polymer crystals.

**Figure 4 polymers-13-03617-f004:**
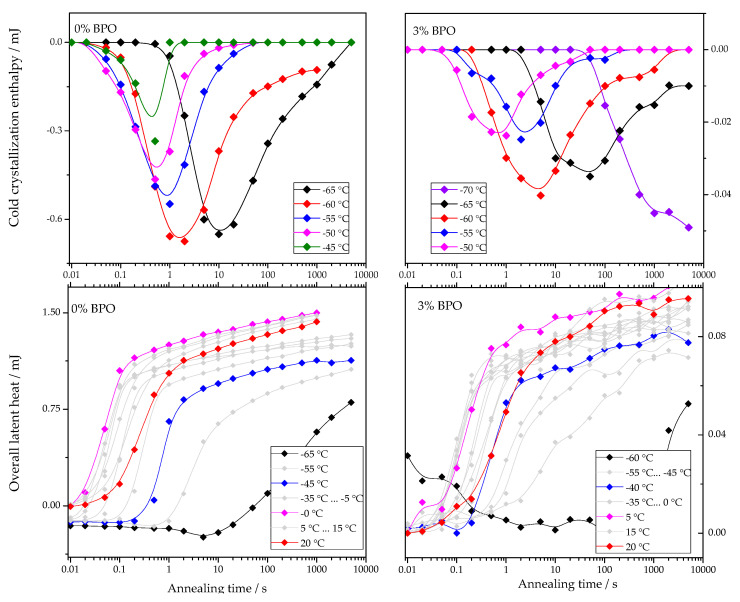
Dependence of cold-crystallization enthalpy and overall latent heat on annealing time and temperature upon heating the cross-linked PCL samples.

**Figure 5 polymers-13-03617-f005:**
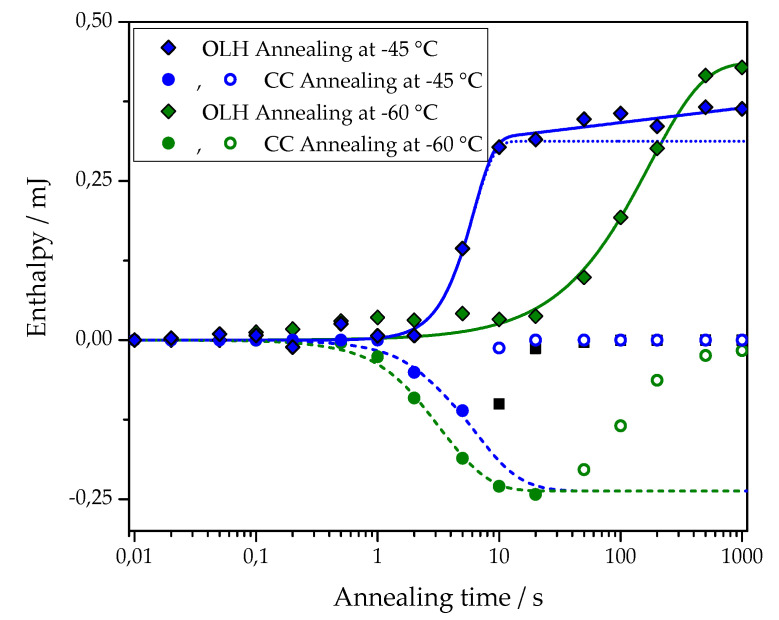
Approximation of the dependence of overall latent heat data (OLH, diamond) on the annealing time with Equation (1) (solid lines). Data for cross-linked PCL with 5% BPO and annealing at −45 °C (blue symbols) and at −60 °C (olive symbols). The dotted line shows the primary crystallization curve, plotted using Avrami equations based on the fit parameters determined for data points obtained after annealing at −45 °C. Approximation of the cold crystallization enthalpy (CC, circles) on the annealing time with Equation (2) (dashed lines). Data for PCL with 5% BPO after annealing at −45 °C (blue symbols) and −60 °C (olive symbols). The fit was performed over the solid points; the hollow points were discarded.

**Figure 6 polymers-13-03617-f006:**
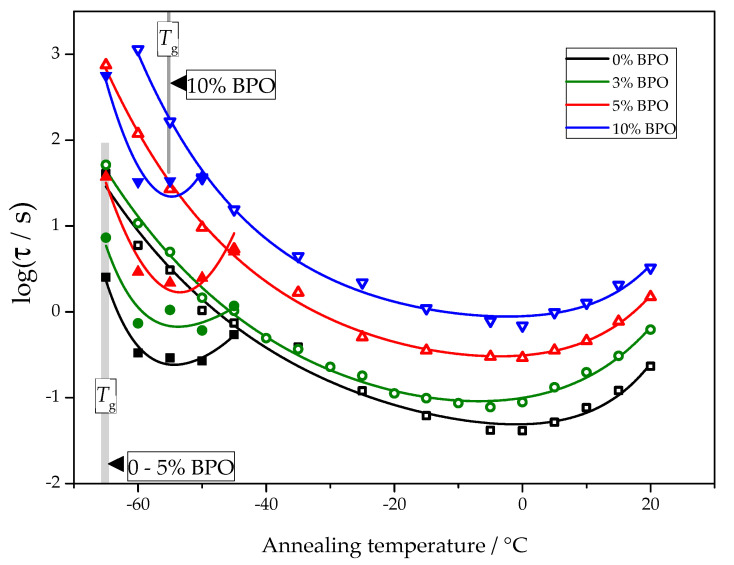
The crystallization halftime (hollow symbols) and nucleation halftime (solid symbols) as a function of the annealing temperature for PCL samples with different cross-link densities.

**Figure 7 polymers-13-03617-f007:**
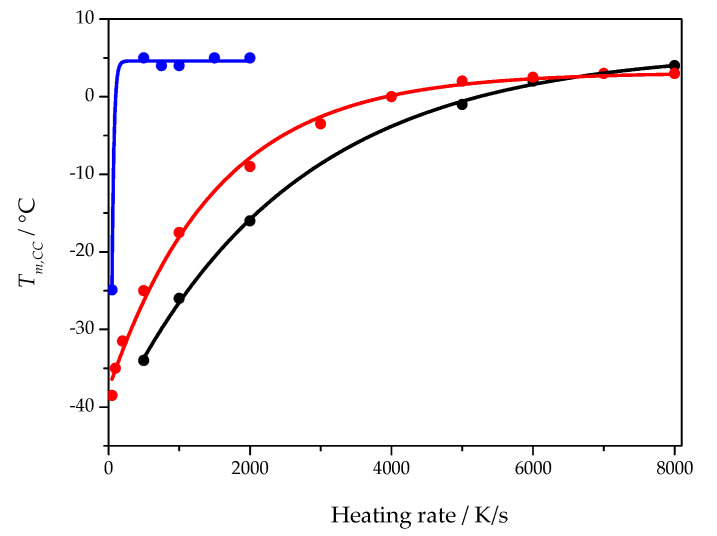
Cold crystallization peak temperature as a function of heating rate for three different samples of cross-linked PCL with 5% BPO. The blue curve was obtained from a sample with little to no nuclei. The black and red curves were obtained from two different samples with a large number of nuclei created before the heating scan.

**Figure 8 polymers-13-03617-f008:**
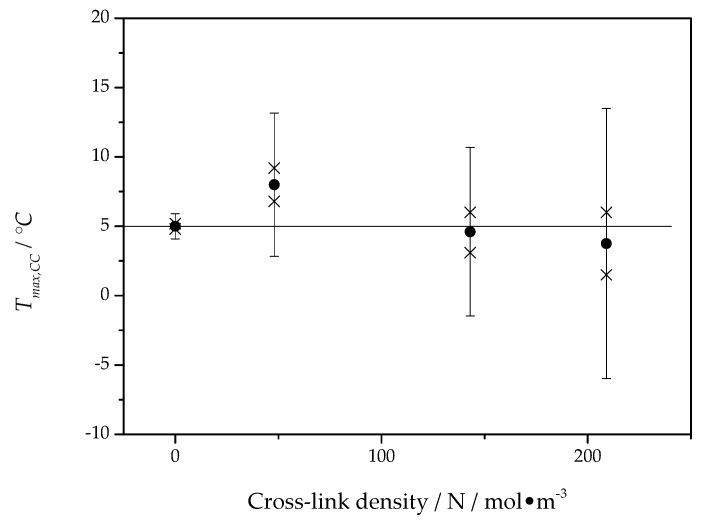
The asymptotic value from fits by Equation (3) to data, as shown in Figure 7, as a function of cross-link density. The horizontal line marks the mean value nearly equal to 5 °C. Crosses represent the the values of $T_{max,CC}^{lim}$ estimated from distinct samples of PCL, solid circles represent the average value of $T_{max,CC}^{lim}$ from different samples, error bars indicate the expanded uncertainty of the estimate.

**Figure 9 polymers-13-03617-f009:**
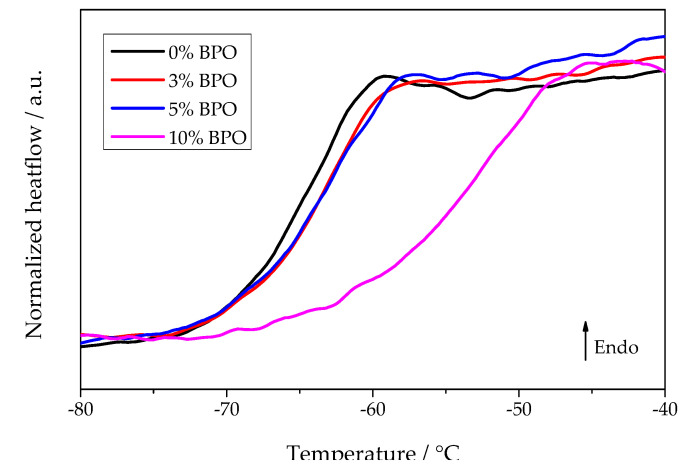
Heating scans of different PCL samples after annealing for 0.1 s at −75 °C. The previous cooling rate was 5000 K/s and the heating rate 1000 K/s.

**Figure 10 polymers-13-03617-f010:**
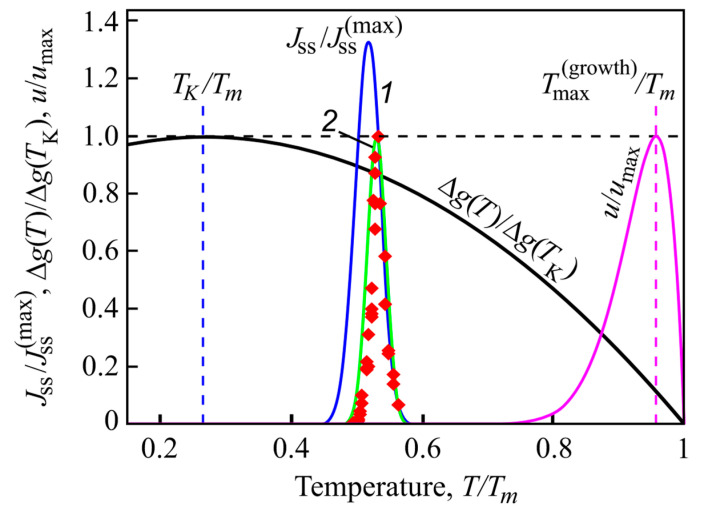
Normalized steady-state nucleation rate, $J_{SS}/J_{SS}^{\left(max\right)}$ and normalized crystal growth rate, *u/u_max_*, in dependence on reduced temperature, *T/T_m_*. Here, $J_{SS}^{\left(max\right)}$ is the maximum nucleation rate and *u_max_* is the maximum growth rate obtained via experiment, *T_m_* is the melting or liquidus temperature. The blue curve (1) shows the theoretical result when the kinetic term in the expression for the nucleation rate is determined via appropriate diffusion coefficients, the green curve (2) is drawn under assumption of validity of the Stokes–Einstein–Eyring equation allowing one to replace the diffusion coefficient by viscosity. Its wide coincidence with experimental data is reached by employing appropriate expressions for the curvature dependence of the surface tension. The reduced thermodynamic driving force, ∆*g*(*T*)*/*∆*g*(*T_K_*), is shown as well, it has a maximum at the Kauzmann temperature, *T_K_*. It is evident that crystallization occurs only in a relatively small temperature range. Typically, the maximum of the growth rate, $T_{max}^{\left(growth\right)}$, is located at temperatures much higher than the maximum of the steady-state nucleation nucleation rate. Reproduced from [35], Figure 1 therein, (2018).

**Figure 11 polymers-13-03617-f011:**
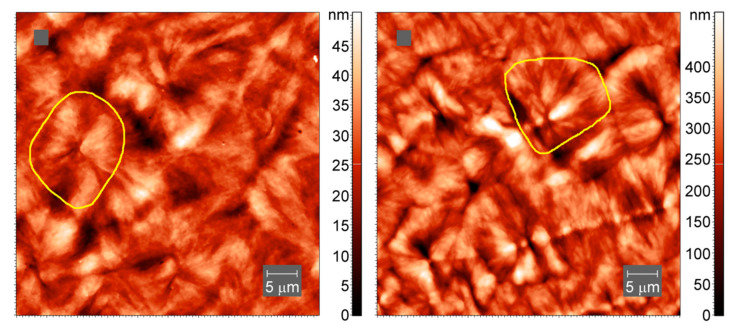
AFM tapping-mode images of spherulites in neat PCL (**left**) and cross-linked PCL with 5% BPO (**right**). Yellow lines outline one spherulite.

**Figure 12 polymers-13-03617-f012:**
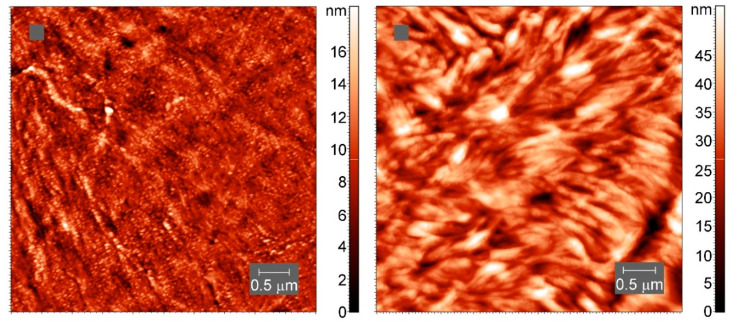
AFM tapping-mode height images of spherulites in neat PCL (**left**) and cross-linked PCL with 5% BPO (**right**). The color indicates the height.

**Table 1 polymers-13-03617-t001:** Spatial cross-link density of the studied samples.

Weight % BPO	*N*/mol·m^−3^
0	0
3	48 ± 4
5	143 ± 4
10	209 ± 3

**Table 2 polymers-13-03617-t002:** The cross-link densities and estimated distances between cross-links for the studied PCL samples.

Sample	*N*/mol·m^−3^	*d*/nm	<*d*>/nm
3% BPO	48 ± 4	3.2	1.8
5% BPO	143 ± 4	2.3	1.3
10% BPO	209 ± 3	2.0	1.1

## Data Availability

The data presented in this study are available on reasonable request from the corresponding authors.

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
