# Peer review of "Crystal Nucleation and Growth in Cross-Linked Poly(ε-caprolactone) (PCL)"

_polymers, 2021, doi:10.3390/polym13213617_

Round 1
Reviewer 1 Report
Mukhametzyanov et al. provided in-depth experimental and theoretical analysis of crystal nucleation and growth in cross-linked PCL. By using fast scanning calorimetry, it was determined that with increasing degree of cross-linking, both the nucleation and crystallization half-times increased. However, the temperatures of the maximum nucleation and the overall crystallization rates remained the same. These observations were then explained by using classical nucleation theory. Both the data analysis and the theoretical interpretation were well explained, thus I recommend the publication of the manuscript after some minor corrections.
1. Several of the figures need to be revised to improve clarity.
-Figure 2, while I understand that the authors may want the readers to focus on the 4 colored curves, the other heating curves shown should also be cleared explained. There annealing time should be indicated.
-Figure 6, figure legend appears to be redundant.
-Figure 7, what’s the difference between the black and red curves
-Figure 8, what do the different symbols represent? And what is the error bar referring to?
2. There are numerous typos throughout the paper, please correct. Also, some of the sentences are convoluted and hard to understand, please revise. For example
-p 9 line 273 “the cross-link density independent position of the max…” this sentence is long and is missing verb
-p 12 line 358 “widely results…”
-p 15 line 460 “enumerator” should be “numerator”
-p 16 line 505 “functional” should be “function”
-etc.
Author Response
The authors express gratitude for reviewers for the help improving the manuscript. The comments from the reviewers are highlighted in yellow. The answers are highlighted in green.
Also, please note that minor changes were made to the references list, to correct for technical errors.
Reviewer #1
- Several of the figures need to be revised to improve clarity.
-Figure 2, while I understand that the authors may want the readers to focus on the 4 colored curves, the other heating curves shown should also be cleared explained. There annealing time should be indicated.
The legend was corrected and the annealing times are shown.
-Figure 6, figure legend appears to be redundant.
Corrected.
-Figure 7, what’s the difference between the black and red curves
The curves were obtained from two distinct samples in the same conditions. The figure caption has been adjusted accordingly.
-Figure 8, what do the different symbols represent? And what is the error bar referring to?
The explanation is now provided in the caption.
- There are numerous typos throughout the paper, please correct. Also, some of the sentences are convoluted and hard to understand, please revise. For example
-p 9 line 273 “the cross-link density independent position of the max…” this sentence is long and is missing verb
Corrected
-p 12 line 358 “widely results…”
Corrected
-p 15 line 460 “enumerator” should be “numerator”
Corrected
-p 16 line 505 “functional” should be “function”
We respectfully disagree, as in that context, it is implied that the degree of crystallization is a function which takes as it’s input another function (of nucleation and growth rates), and thus mathematically is a functional. As noted in the sentence such treatment of the degree of crystallization as a functional and not as a function is an essential ingredient of a correct description advanced in the present paper.
-etc.
Reviewer 2 Report
On request of Polymers, I have revised the manuscript titled “Crystal nucleation and growth in cross-linked poly(ε-caprolactone) (PCL)”, by Timur Mukhametzyanov et al.
With their study, the authors have experimentally investigated the crystal nucleation and overall crystallization kinetics of cross-linked poly(ε-caprolactone), by fast scanning calorimetry technique in a wide temperature range. Appealing results and interesting correlations between the degree of cross-linking and the nucleation, crystallization half-times, as well as glass transitions were found and reported. No correlation was instead detected between the degree of cross-linking and the temperatures of the maximum nucleation and the overall crystallization rates. Moreover, a theoretical interpretation of the reported results has been provided in terms of classical nucleation theory.
General Comments
Poly(ε-caprolactone) (PCL) is a biocompatible and biodegradable industrial polymer with production output amounting to tens of thousands of tons every year. Considering the range of current and possible applications of PCL, it is necessary modifying its chemical structure by creating cross-links between polymer chains of PCL, by adopting suitable cross-linking agents, to have greater mechanical strength and a two-way shape memory effect which is not exhibited by the pristine PCL.
In this regard, knowing how the crosslinking influences the crystallization behaviour of the polymer under different conditions is mandatory. Therefore, the present study, which is also an extension of a previous study by the same authors, could be considered welcome. By containing an analysis of PCL crystallization at isothermal conditions, and an investigation of the effect of cross-link density on crystal nucleation and crystallization rates of PCL, as well as the glass transition temperature and relaxation kinetics, this manuscript provide the scientific community with relevant information. In addition, which factors are the basic ones determining the effect of cross-linking on crystallization in the polymer analysed were unveiled.
The topic of the present manuscript appears original, interesting, and meets the requests of the several sectors. I think that it could attract the interest of a wide audience. The English is good and fine.
Only some minor issues, concerning the organization of the contents should be addressed before approving the publication on Polymers.
Several experimental details or explanations which (in my opinion) should be included either in the Material and Methods section or in the main text (in the more appropriate section) have been included in the Figure captions, which appear extremely long. Therefore, I suggest moving experimental details or explanations reported in some Figure captions (Figure 5, 7, 10) in the main text.
Line 119. Please, add the missing round bracket.
Section 4. The titles of the sub-sections, i.e., 4.1, 4.2, etc. do not respect the template provided by Polymers. Please, check the template and correct accordingly.
Conclusion are poor and must be extended.
Author Response
Reviewer #2
Several experimental details or explanations which (in my opinion) should be included either in the Material and Methods section or in the main text (in the more appropriate section) have been included in the Figure captions, which appear extremely long. Therefore, I suggest moving experimental details or explanations reported in some Figure captions (Figure 5, 7, 10) in the main text.
Corrected, except for Figure 10 where we prefer to keep the descriptions of the models in the figure caption. In the text, we replaced (see Figure 10) by „Typical locations of the maximum of the nucleation and growth rates confirming this statement are shown in Figure 10“.
Line 119. Please, add the missing round bracket.
Corrected
Section 4. The titles of the sub-sections, i.e., 4.1, 4.2, etc. do not respect the template provided by Polymers. Please, check the template and correct accordingly.
Corrected
Conclusion are poor and must be extended.
We have expanded the conclusions sections to better represent the results presented in the manuscript.
